# Orally Dispersible Swallowed Topical Corticosteroids in Eosinophilic Esophagitis: A Paradigm Shift in the Management of Esophageal Inflammation

**DOI:** 10.3390/pharmaceutics17101325

**Published:** 2025-10-13

**Authors:** Alberto Barchi, Marina Girelli, Antonio Ventimiglia, Francesco Vito Mandarino, Silvio Danese, Sandro Passaretti, Mona-Rita Yacoub, Serena Nannipieri, Ambra Federica Ciliberto, Luca Albarello, Alessandra Bartolucci, Edoardo Vespa, Giuseppe Dell’Anna

**Affiliations:** 1Gastroenterology and Digestive Endoscopy, IRCCS Ospedale San Raffaele, Via Olgettina 60, 20132 Milan, Italy; barchi.alberto@hsr.it (A.B.); girelli.marina@hsr.it (M.G.); ventimiglia.antonio@hsr.it (A.V.);; 2Unit of Immunology, Rheumatology, Allergy and Rare Diseases, IRCCS San Raffaele Scientific Institute, 20132 Milan, Italy; 3Pathology Unit, IRCCS San Raffaele Scientific Institute, 20132 Milan, Italy

**Keywords:** eosinophilic esophagitis, corticosteroids, orally dispersible, formulations, topical treatment

## Abstract

Eosinophilic esophagitis (EoE) is a chronic, immune-mediated disease of the esophagus within the type 2 inflammatory spectrum, characterized by progressive tissue remodeling driven by uncontrolled inflammation. Its incidence and prevalence are rising sharply, likely reflecting environmental triggers acting on genetic and epigenetic susceptibility. Therapeutic options have expanded rapidly, with recent approvals of new topical steroidal formulations together with biologic compounds. Proton pump inhibitors (PPIs), older swallowed topical corticosteroid (STC), and dietary interventions remain in use but are limited by suboptimal adherence and treatment discontinuation. This has driven a shift toward advanced orally dispersible STCs formulations—most notably budesonide orally dispersible tablets (BOT), budesonide oral suspension (BOS), and fluticasone orally dispersible tablets (FOT). BOT, the most extensively studied, achieves high rates of histologic and clinical remission, with favorable safety and superior adherence compared to earlier STCs formulations. This comprehensive overview focuses on following key research findings and novelty aspects of new treatments: (a) optimized esophageal targeting through orally dispersible or viscous formulations of STC, enhancing mucosal contact time and improving drug delivery to affected tissues compared to older formulations; (b) robust evidence for both induction and maintenance rates of remission, with data extending up to nearly 2 years and showing stable efficacy across clinical, histologic, and endoscopic endpoints; (c) effectiveness in STC-refractory patients, with BOT showing benefit even in those previously unresponsive to older STC formulations. This review synthesizes evidence of steroid therapy in EoE, from pharmacological aspects to clinical efficacy from randomized trials and emerging real-world studies, highlighting its impact on EoE management and outlining future therapeutic directions.

## 1. Introduction

Eosinophilic esophagitis (EoE) is a chronic immune-mediated disease of the esophagus [1,2]. It is characterized by eosinophilic infiltration of the esophageal mucosa, leading to the development of obstructive symptoms [3], after having ruled out other possible causes of eosinophilic infiltration of the esophagus [4]. EoE incidence and prevalence have been steadily increasing in the last 20 years, especially in Europe and North America [5]. Recently, its prevalence has been estimated to be around 40.1 per 100,000 inhabitants-years and 5.3 new cases per 100,000 inhabitants per year [6]. There is a male-to-female ratio of 3:1, with a higher incidence in adults compared to children [1]. The typical age of onset is around 25–35 years old [7]. EoE leads to esophageal remodeling with lumen narrowing and stenosis development, if untreated [8]. EoE symptoms are obstructive in the adult population (mainly dysphagia and food impaction) while poor appetite, abdominal pain, vomiting, and heartburn/regurgitation are more common in the pediatric population [9]. Furthermore, the chronic nature of the disease might negatively impact patients’ quality of life, causing social isolation, lifestyle adjustments, food avoidance, and compensatory behaviors [10]. The gold standard for the diagnosis remains upper endoscopy (EGD) with biopsy sampling from different sites of the esophagus, being EoE a patchy disease [11]. According to current guidelines, at least six samples should be collected from at least two esophageal regions [11], with ≥15 eosinophils per high-power field (HPF) of peak eosinophil count (PEC) as the histologic cut-off for the diagnosis [11]. The treatment landscape in EoE is rapidly evolving: (a) Proton Pump Inhibitors (PPIs) remain widely used as off-label drugs, with efficacy rates >50% in recent meta-analysis even after dose-escalation [12,13]. Even though they are of limited efficacy eosinophilic infiltration is high, and are burdened by high rates of discontinuation or withdrawal rates [14], they seem effective especially in selected EoE phenotypes (inflammatory profile with prevalent distal esophageal eosinophilia [15]) at twice daily dose [16]; (b) elimination diets (1-, 4-, 6-foods or elemental diets) appear to be effective [17], even for less stringent approaches [17], but are burdensome for patients and it is difficult to select potential responders [18]; (c) biological treatments are on the pipeline, leaded by dupilumab, already proven as the most effective therapy, especially for refractory cases [19], while several other monoclonal antibodies are being investigated, with several different targets (IL-4, IL-5, IL-13, IL-15, and Siglec-8 pathways) [20,21,22] and even immunomodulators have shown anti-eosinophil promising effects [23]. For fibrostenotic disease, esophageal mechanical or hydraulic dilation represents a fundamental additional therapeutic option [24]. Nonetheless, swallowed topical corticosteroids (STCs), fluticasone and budesonide above all, still represent the cornerstone of EoE treatment [2]. For years, STCs have been used in formulations and dosages borrowed from other settings and used off-label, which raises several issues regarding adherence and patient compliance [25]. Recently, novel formulations and compounds have been developed, completely re-shaping EoE treatment and the natural history of the disease (Figure 1) [26]. Despite that, several unmet needs remain unaddressed in EoE management, particularly regarding treatment adherence, drug formulations, and strategies for long-term disease control. This situation posed economic, logistical, and acceptability barriers, which negatively impact therapeutic adherence. Moreover, most patients tend to relapse after therapy discontinuation, highlighting the need for safe and effective long-term maintenance strategies. The recent introduction of targeted formulations (e.g., orally dispersible tablets or ready-to-use suspensions, already approved in several countries) as well as specific biologic therapies has helped to overcome some of these obstacles; however, issues remain regarding ease of administration, insurance coverage/reimbursement, long-term safety evidence, and personalized strategies to improve adherence and sustain remission [27]. The aim of this comprehensive review was to exploit the change in paradigm that the novel formulations of STCs have brought to the EoE treatment landscape by conducting an extensive review of the literature alongside a detailed comparative analysis of pharmacokinetic properties.

## 2. Off-Label (ol)-STCs Formulations

OlSTCs formulations (Figure 2), mainly swallowed aerosolized fluticasone (SAF) or swallowed nasal drops suspension (FNDS), have been initially translated from the allergology/immunology pediatric setting, and investigated with good results in terms of induction of remission, compared to placebo [28], and to other off-label treatments (mainly PPIs) [29,30]. Early studies highlighted controversies regarding the real-life efficacy of SAF/FNDS compounds, particularly in terms of clinical outcomes in the adult population [31], and especially after dose de-escalation for maintaining remission [13]. In a pivotal pediatric study following a strict regular follow-up schedule, Andreae et al. demonstrated the effectiveness of SAF in inducing and maintaining remission in EoE, without serious adverse events in the long-term, in terms of either oro-esophageal candidiasis or adrenocortical down-regulation effects [32]. Nonetheless, the largest study to date on real-world utilization and efficacy of olSTCs from the EoECONNECT registry [33] clearly highlighted how fluticasone-based therapies for EoE were heterogeneous in (a) method of administration, (b) dose regimens ranging from 0.4 mg/day to 1.6 mg/day. The authors pointed out how the efficacy rates, both in terms of histologic, clinical, and clinico-histological remission, were sub-optimal and dose-dependent (spanning from 41% for lower doses to 71.4% at higher doses) [33]. Mometasone furoate is a less commonly used STCs compound [34], with 10 times lower bioavailability compared to fluticasone and more than 300 times lower than budesonide [34]. Notably, mometasone is more lipophilic compared to the other corticosteroids, with greater mucosal adhesion and higher affinity for the steroid receptor [34]. Aerosolized mometasone administered in metered-dose inhalers (MDIs) with a posology of 200 μg four times a day showed a fairly good clinical response in terms of dysphagia symptoms in pediatric EoE patients in a two-month observational prospective study [35]. Syverson et al. reported data on mometasone furoate (with dosages up to 1500 μg per day, weight-adjusted) in a pediatric cohort of EoE patients, with a compound of viscous suspension composed of mometasone powder combined with methylcellulose solution and additive/flavoring agents [36]. This compound demonstrated optimal outcomes in terms of median eos HPF change from baseline (−50, *p* < 0.001), with non-significant differences between steroid-failure and steroid-naïve patients (*p* = 0.11) [36]. Aerosolized mometasone via MDIs was investigated in a placebo-controlled eight-week induction RCT on 36 EoE patients, demonstrating statistically significant improvement in dysphagia scores (*p* < 0.01); nevertheless, histological or endoscopic outcomes were not assessed [37]. Recently, mometasone was rediscovered in a novel esophageal-targeted formulation, thereby increasing interest in this less-investigated molecule [38,39].

Budesonide, as an active principle for EoE, has spurt after fluticasone, as indicated from recent real-world data, reporting that most patients were first prescribed fluticasone and only later were shifted towards budesonide compounds [33].

Furthermore, budesonide compounds have shown high rates of clinico-histological efficacy since the first trials, with less heterogeneity in delivery methods [33]. Dohil et al. firstly investigated the effectiveness of an Oral Viscous Budesonide (OVB) (Pulmicort Respule^®^ 0.25 mg/mL, already approved for bronchial asthma) reduction in PEC, which evidenced a significant reduction in PEC compared to placebo (*p* < 0.0001) in a pediatric setting [40]. Another RCT confirmed these results in a short-term (15 weeks) induction period, at a 2 mg daily dosage [41]. These studies investigated an OVB formulation, in which budesonide nebulized suspension was mixed with a viscous agent (mainly sucralose) to prolong mucosal contact time and was then swallowed to coat the esophagus [33]. These compositions of OVB or “*slurry budesonide*” have been the most investigated, since, differently from fluticasone compounds, budesonide has hardly been studied as an aerosolized formulation delivered with MDIs [33]. These studies have explored at least 2 OVB dosages (1 mg vs. 2 mg twice a day) with no difference in terms of clinic-histologic remission rates. Straumann and colleagues evaluated the efficacy of OVB also in the long-term period (1 year), at a minimum dosage of 0.25 mg twice a day, compared to treatment withdrawal (placebo), reporting higher relapse rates in the placebo group [42]. Multiple real-world studies confirmed the efficacy of OVB in the maintenance of remission in EoE, even with strict dose de-escalation protocols, both in children [43] and adults [44,45]. In a landmark retrospective study, Greuter and colleagues investigated composite remission during the maintenance period of 2.2 years with OVB either at low or high dose (0.5 mg vs. 1 mg twice a day), finding no significant differences in terms of relapse rates, which were nonetheless considerably high (72% vs. 54%, 67% overall) [45]. Lower dosages were related to earlier relapses. More recent data, with longer follow-up, have attempted to explore the main causes of these high relapse rates with olSTCs [46,47]. McCallen and colleagues identified lower rates of clinical and histological remission in poor adherence compared to optimal adherence group in a >6.5 years of follow-up [46]; while Reed et al. recently did not identify significant impact of daily vs. twice a day dosing on the odds of histologic response at a multivariable analysis (adjusted odds ratio, 1.03; 95% confidence interval, 0.67–1.60) [47]. The sub-optimal rates of clinico-histological remission at induction [33], the high relapse rates, and the conflicting results on histologic, clinical, and endoscopic outcomes during the maintenance period [13], warranted closer inspection of causal factors impacting on olSTC effectiveness. The active principle seems not to represent an issue since Dellon et al. demonstrated no significant differences in terms of endoscopic, clinical, and histological endpoints comparing budesonide (OVB) and fluticasone (SAF) in a randomized setting [48]. Non-intuitive administration methods or home-made preparation by the patient are surely major factors playing a role in worse efficacy outcomes of olSTCs, as demonstrated by real-world analysis [33]. For example, fluticasone propionate, administered via swallowed nasal drops, has shown higher efficacy compared to other fluticasone formulations, while the novel budesonide orally dispersible tablets (BOT) have proved the most effective, with this novel compound increasing drug mucosal contact and higher patient compliance [33]. Furthermore, recently different viscous agents, such as hydroxyethyl cellulose, have been added to OVB formulations to increase the adhesion strength of the preparation [49]. More specifically, viscous slurry compounds (mainly budesonide) have been proven more effective than nebulized inhalers in terms of eosinophil reduction in a randomized clinical trial [50]. Dellon and colleagues also demonstrated a higher mucosal contact time for the OVB group than the nebulized budesonide (*p* < 0.005), as measured by scintigraphy [50]. Dose ranges surely play a major part in defining the sub-optimal efficiency of olSTCs. Both SAF and OVB have shown increasing clinico-histological remission rates with a dose-dependent relation (≥0.8 mg/day for fluticasone and >1 mg/day for budesonide) [33]. Concerning posology, pediatric data showed that OVB administered every other day is not effective and potentially could lead to higher relapse rates [51]. Also, patient-related or disease-specific factors are related to STCs responsiveness. Greuter et al. observed that although women are less frequently affected by EoE, they appear to demonstrate a higher response rate to treatment with topical corticosteroids. Furthermore, patients who experienced a shorter diagnostic delay were more likely to maintain remission even after discontinuation of STC therapy [52]. This indicates that the duration of the untreated pre-diagnostic phase may significantly impact the therapeutic response to topical corticosteroids [52]. This can be explained by the fact that prolonged diagnostic delays increase the likelihood of developing fibrotic complications associated with EoE—such as esophageal rings and strictures—detected during EGD. These fibrotic changes are generally more resistant to treatment compared to the inflammatory features typical of early-stage EoE. Therefore, early diagnosis and timely initiation of appropriate therapy are essential for optimal disease management, as has also been demonstrated in other chronic inflammatory gastrointestinal disorders, including inflammatory bowel disease [2]. The question marks raised by olSTCs on therapeutic adherence, variability in preparation, dosages, and ease of use have led to the development of new standardized formulations such as orally dispersible tablets (ODTs) and ready-to-use suspensions, marking a substantial advancement, as outlined in Table 1. In the multivariable analysis from the EoE CONNECT registry, ODTs (particularly those with budesonide as the active principle) emerged as the only formulation significantly associated with an increased odds of remission (OR 18.9, *p* < 0.001), regardless of age, sex, or disease phenotype [33]. Even during the maintenance phase, after dose-reduction, the novel formulations demonstrated greater stability, with only 46% of patients treated with fluticasone-based SAF/FNDS in stable remission, compared to over 69% for those treated with budesonide (in all its forms), but with BOT especially [33]. The introduction of modern STCs formulations optimized for esophageal delivery has therefore addressed many of the limitations associated with older therapies, improving both clinic-histological efficacy and patient adherence [4]. The routine use of standardized preparations such as BOT now represents the most effective and reproducible topical treatment option available for EoE [53].

## 3. Budesonide (BOT) and Fluticasone (FOT) Orally Dispersible Tablets

BOT represents a major innovation in the therapeutic management of EoE. This formulation is the first pharmacological agent specifically engineered with targeted delivery to the esophagus. BOT received approval from the European Medicines Agency (EMA) in January 2018 for use in the induction of remission in adult patients with EoE, and it is currently available in several European countries [54]. BOT formulation was explicitly designed to optimize topical corticosteroid delivery to the esophageal mucosa through orally dispersible technology, which enhances mucosal adhesion and prolongs local drug exposure [55]. BOT efficacy in the induction of remission has been validated in several randomized controlled trials (Table 2). A Phase 2 RCT (EOS-2) explored the efficacy of an early formulation of BOT (in two different dosages: 2 mg daily and 2 mg twice a day) compared to placebo and OVB. The authors found comparable outcomes in histological, endoscopic, and clinical remission for all 3 budesonide groups compared to placebo [56]. Notably, tolerance and satisfaction of patients were higher in the BOT group compared to the OVB. These reassuring results led to the development and investigation of orally dispersible tablets as novel administration methods. Lucendo et al. showed that 6 weeks of treatment with 1 mg of BOT twice daily achieved complete clinic-histological remission, defined as dysphagia and odynophagia severity scores ≤ 2 on a 0–10 scale for each of the seven days preceding the end of the blinded phase, and a PEC < 5 eos/hpf in 58% of patients, compared to 0% of placebo [55]. Histological remission was seen in 93% of patients who received BOT compared to 0% in the placebo arm. Moreover, the authors showed that extending treatment to 12 weeks increased the cumulative clinico-histological remission rate to 84.7%, underscoring the value of prolonged induction in improving treatment outcomes [55].

Miehlke et al., in an open-label phase 3 randomized-controlled trial complementary to the *EOS-2* Program, reported clinico-histological remission rates of 69.6% within six weeks of therapy with BOT 1 mg twice a day [57]. The authors demonstrated significant reductions in mean PEC as well as patient-reported symptom scores (DSQ) and endoscopic findings [57]. Furthermore, BOT was well tolerated, with the most common adverse event being mild oropharyngeal candidiasis, which could be easily managed with antifungal therapy [55,57]. A post hoc analysis evaluated whether BOT could achieve “deep remission” according to the stringent composite criteria defined by Greuter et al. [52], which included complete resolution of symptoms (deep clinical remission), absence of eosinophils on histology (deep histologic remission), and normalization of endoscopic appearance (deep endoscopic remission). A significant proportion of patients met all these criteria as early as six weeks into therapy [58]. Recent systematic reviews and meta-analyses have provided further evidence of the efficacy of BOT. Rawla and colleagues found that budesonide markedly increased the odds of histological remission compared to placebo, with a pooled risk ratio (RR) of 11.93 (95% CI, 4.82–29.50), along with significant reductions in eosinophil counts and symptomatic improvement [59]. Visaggi et al. in their network meta-analysis, ranked BOT 1 mg twice a day as the most effective treatment for histological remission (≤15/≤6 eos/hpf), resolution of symptoms, and endoscopic improvement [60]. The long-term efficacy of BOT has also been proven. In a 48-week RCT, Straumann et al. showed that maintenance therapy with BOT at both 0.5 mg and 1.0 mg BID achieved sustained remission in 73.5% and 75% of patients, respectively, far superior to placebo, where the remission rate was only 4.4% [61]. Both dosing regimens were well-tolerated, and adverse effects were rare. Deep endoscopic remission was maintained in 79.2% and 71.1% of patients on BOT 0.5 mg and 1.0 mg BID, respectively, against just 8.5% in the placebo group. It also led to significant and durable improvements in health-related quality-of-life measures as assessed by the validated EoE-QoL-A instrument [61]. Long-term outcomes were further assessed in a 96-week open-label extension (OLE) study by Biedermann et al. [62], which involved patients who had completed the preceding 48-week double-blind phase with either 1/0.5 mg twice daily of BOT. The study confirmed sustained clinico-histological remission in the majority of participants, with 81.9% and 80.1% of sustained clinical and histological remission (≤15 eos/HPF) at week 96, respectively [62]. Even at a more stringent histologic cut-off (≤6 eos HPF) rate of remission remained high (77.8%). These results included cohorts of patients undergoing continuous BOT treatment and those undergoing six weeks of reinduction (BOT 2 mg daily) at the OLE baseline. Notably, when analyzing only 93 patients undergoing continuous BOT therapy without re-induction, at 96 weeks, 86% of patients remained in clinical remission, with optimal outcomes in terms of quality of life and endoscopic appearance. No additional safety issues were identified during the extended treatment period [62]. It is worth mentioning, nonetheless, that over 15% of patients relapsed at the end of the study period, stressing the relevance of continuous clinical and endoscopic follow-up even in patients achieving deep remission [62]. Recent pooled data did not identify dose tapering of BOT in the maintenance phase as a cause of increased risk of histologic relapse (RR 1.04; 95% CI: 0.72–1.51), indicating that gradual dose reduction can be an alternative strategy for certain patients [13]. However, the relevant risk of histologic relapses with reduced maintenance dose regimens underscores the importance of personalized treatment planning and rigorous monitoring during maintenance therapy [63].

**Table 2 pharmaceutics-17-01325-t002:** Summary of main outcomes of studies on Budesonide Orally Dispersible Tablets (BOT).

Study (First Author, Year)	Study Design	Intervention	Duration	Population (Age)	Primary Outcome	Key Results
Lucendo et al. (2019) [55]	Phase 3, RCT, DB, PC (*EOS-1*)	BOT 1 mg BID vs. Placebo	6–12 weeks	88 adults (18–75 years)	**Complete CHR:** <16 eos/mm^2^ HPF (<5 eos HPF) Symptom severity ≤2 points on NRS scale	Complete remission: 58% BOT vs. 0% Placebo; Histologic remission: 93% BOT vs. 0% Placebo (*p* < 0.0001)
Miehlke et al. (2016) [56]	Phase 2, RCT, DB, PC	BOT 2 mg QD/2 mg BID/OVB 2 mg QD vs. Placebo	2 weeks	77 adults (18–75 years)	**Histological remission:** <16 eos/mm^2^ HPF (<5 eos HPF) Change in mean PEC (eos/mm^2^ HPF)	Histological response (<65 eos/mm^2^ HPF/<20 eos HPF) in 100% and 94.7% for both BOT dosages. 0% in placebo Histological remission was 84.2% and 89.5% compared to 73.7% in OVB Higher tolerance and satisfaction for BOT compared to OVB
Miehlke et al. (2021) [57]	Phase 3, open-label induction for RCT, DB, PC (*EOS-2*)	BOT 1 mg BID	6 weeks	181 patients (18–75 years)	**Complete CHR:** <16 eos/mm^2^ HPF (<5 eos HPF) Symptom severity ≤2 points on NRS scale	-CHR 69.6% Histological remission: 90.1% (deep remission 0 eos HPF 84.5%) Clinical remission 75.1%. Significant endoscopic improvement (*p* < 0.0001)
Straumann et al. (2020) [61]	Phase 3 maintenance RCT, DB, PC (*EOS-2*)	BOT 0.5/1.0 mg BID vs. Placebo	48 weeks	204 patients (18–75 years)	**Maintenance of remission:** n° of pts not in clinical relapse (≥4 points on NRS scale) n° of pts not in histologic relapse (≥48 eos/mm^2^ HPF/≥15 eos HPF)	Maintained remission: 73.5% (0.5 mg BID), 75% (1.0 mg BID), 4.4% (Placebo)
Biedermann et al. (2025) [62]	OLE of a randomized, DB, PC, 48-week maintenance trial (*EOS-2*)	BOT 0.5 or 1.0 mg BID or 2.0 mg BID for OLRI	96 weeks	186 patients (extension of previous RCT) (18–75 years)	**Maintenance of remission:** n° of pts not in clinical relapse (≥4 points on NRS scale) n° of pts not in histologic relapse (≥48 eos/mm^2^ HPF/≥15 eos HPF)	Clinical remission: 81.9% Histological remission: 80.1% (deep remission 0 eos HPF 78.8%) CHR 78.1% Histological relapses 15.1% EREFS stable through week 48 to 96 High patient satisfaction

OLE—Open-label extension; OVB—Oral Viscous Budesonide; BOT—Budesonide Orally Dispersible Tablets; BOS—Budesonide Oral Suspension; RCT—Randomized Controlled Trial; DB—Double-Blind; PC—Placebo-Controlled; CHR—Clinico-Histologic Remission; BID—Bis in Die; QD—Once a day; HPF—High Power Field; QoL—Quality of Life; EREFS—Eosinophilic Esophagitis Endoscopic Reference Score; EoE—Eosinophilic Esophagitis; PEC—Peak Eosinophil Count; OLRI—Open-Label Re-Induction.

In conclusion, BOT offers a number of advantages, including enhanced esophageal targeting, favorable tolerability, and high patient adherence due to its ease of administration. It has become a central component of pharmacologic therapy for EoE, with robust evidence supporting its efficacy for both induction and maintenance of remission. Clinical trials consistently demonstrate their superiority over placebo, and their safety profile remains reassuring across both short- and long-term treatment horizons. While further research is needed to clarify the optimal dosing strategy, duration of maintenance therapy, and safe discontinuation protocols, current evidence strongly supports the use of BOT as a first-line treatment in EoE. As demonstrated by early studies on old formulations, given its topical mechanism of action and low systemic bioavailability, fluticasone is well-suited for targeting esophageal inflammation while minimizing systemic side effects [33]. As abovementioned, old fluticasone compounds were adapted from asthma therapy, with patients instructed to swallow nebulized suspensions or dry powders via MDIs to deliver the steroid to the esophagus [64]. This approach was effective in reducing eosinophilic counts and symptoms but was hampered by variability in esophageal contact time and poor palatability, which affected treatment adherence [65]. The advent of *APT-1011*, a fluticasone orally disintegrating tablet (FOT), represents a major innovation and an additional weapon for EoE clinicians [66]. Unlike swallowed inhalers (the already mentioned SAF/FNDS via MDI), *APT-1011* is an ODT designed specifically for esophageal delivery: it dissolves on the tongue, has to be swallowed without water, and maximizes mucosal coating [66]. This enhances exposure to the esophageal epithelium and offers consistent drug delivery. The *FLUTE* phase 1/2b randomized trial demonstrated that *APT-1011* significantly improved histologic remission, endoscopic features (especially rings and linear furrows), and symptom scores at 8 weeks of follow-up [66]. Particularly, the 1.5 mg twice daily regimen improved the median PEC in both the distal and proximal esophagus (92% and 100%, respectively) compared to the 3 mg once-daily regimen at bedtime (66% and 97%, respectively) [66]. Once-daily administration nonetheless promoted better compliance, an important consideration in chronic diseases [66]. Additional outcomes were retrieved from phase 3 of the *FLUTE* trial, where four different FOT dosages were analyzed: 3 mg twice daily, 3 mg once daily at bedtime (HS), 1.5 mg twice daily, and 1.5 mg at bedtime [67]. Histological remission rates (≤6 eos HPF) at the first timepoint assessment (12 weeks) were comparable to those in the phase 2 trial (higher for 1.5 mg twice daily, 86%, and 3 mg twice daily, 80%) with significance for all doses versus placebo (*p* < 0.001) [67]. These significant response rates were maintained at 26 and 52 weeks [67]. Dysphagia scores also improved consistently throughout the study period, while a post hoc analysis showed fair to good improvement of fibrostenotic endoscopic features [67]. Fluticasone reduces inflammation by inhibiting eosinophil recruitment and survival, suppressing pro-inflammatory cytokines (e.g., IL-5, IL-13) (Figure 1), promoting tissue healing, and slowing the progression of fibrostenosis [68]. Its topical effect is localized, with minimal systemic absorption due to extensive first-pass hepatic metabolism, limiting risks such as adrenal suppression [69]. The main adverse event observed with fluticasone use in EoE is oropharyngeal and esophageal candidiasis, especially with twice-daily regimens. This risk is lower in once-daily dosing and is typically manageable with antifungal therapy or dose adjustment. No significant systemic corticosteroid side effects were observed in trials using *APT-1011* [67]. In conclusion, fluticasone has transitioned from an improvised swallowed inhaler to a targeted, disease-specific therapy [66]. Its role in EoE is now central, offering both induction and maintenance of remission, and FDA/EMA approval is eagerly awaited. With the support of robust clinical evidence, particularly thanks to the FLUTE trial, FOT formulations are likely to become the standard of care, bridging the gap until biologic therapies become widely available [67]. Larger multicenter trials are warranted to confirm optimal dosing strategies.

## 4. Budesonide Oral Suspension (BOS)

Budesonide oral suspension (BOS) is a viscous, muco-adherent corticosteroid preparation, which has been developed specifically for esophageal delivery [70]. BOS received Food and Drug Administration (FDA) approval in February 2024. It was designed to maximize topical anti-inflammatory effects in the esophagus by increasing mucosal contact time and minimizing systemic exposure [70]. Its viscous formulation allows the medication to coat the esophageal mucosa effectively, ensuring that the active corticosteroid remains in prolonged contact with the inflamed tissue, thus enhancing local efficacy while limiting systemic side effects [70]. The pharmacokinetic profile of BOS has been characterized in pediatric populations through a phase 2 study involving children and adolescents (ages 2–18 years) [70]. After age and weight-based dose adjustments, peak plasma budesonide concentrations were achieved approximately one hour post-administration, with a terminal half-life of approximately 3.3 to 3.5 h [70]. Importantly, systemic exposure remained consistent across age groups when appropriate dosing adjustments were made, supporting the suitability of BOS across a wide pediatric age range. These pharmacokinetic features also support the safety of BOS even for long-term administration [70]. This formulation was engineered to be different from the already known OVB (*slurry* compound) [40,41]. BOS is a suspension with a combination of two viscosity-modifying agents giving it a syrup-like consistency (as opposed to a slurry), along with flavoring agents and preservatives [71]. Several RCTs have established the efficacy of BOS in achieving both histological and clinical remission in patients with EoE (Table 3). Gupta and colleagues firstly investigated BOS in children aged 2–18 years, reporting significant improvement in histological (≤6 eos HPF) and clinical (EoE symptom score reduction >50%) for high (47.1%, *p* = 0.0174) and medium doses (52.6%, *p* = 0.0092) compared to placebo (5.6%) [71]. Interestingly, no significant response rate was identified for low dose (*p* = 0.5282) [71]. However, improvements in symptom scores were not significantly different across groups, likely reflecting a high placebo effect frequently observed in pediatric trials [72]. In a pivotal phase 2 trial, 93 adolescent and adult patients (ages 11–40 years) were randomized to receive BOS 2 mg twice daily or placebo for 12 weeks. The BOS group exhibited a significantly higher histologic response rate (≤6 eos HPF) compared to placebo (39% vs. 3%; *p* < 0.0001) [73]. Symptom improvement, assessed through DSQ, was also superior in the BOS group (−14.3 vs. −7.5; *p* = 0.0096). Endoscopic findings improved significantly in BOS-treated patients, as measured by EREFS, further validating endoscopic healing [73]. Building upon these results, a phase 3 trial (ORBIT1/SHP621-301) evaluated 318 patients aged 11–55 years, randomized 2:1 to BOS 2 mg twice daily or placebo for 12 weeks. The study confirmed the superiority of BOS, with 53.5% of patients achieving histological remission, compared to only 1% in the placebo group (*p* < 0.001). Moreover, symptom response (defined as ≥30% DSQ improvement) was achieved in 52.6% of BOS patients compared to 39.1% in the placebo group (*p* = 0.024). These results confirmed BOS as an effective agent for both objective histologic healing and symptomatic relief in EoE [74]. While eosinophil count reduction remains a primary endpoint in EoE studies, other histologic features such as basal zone hyperplasia (BZH), eosinophilic micro abscesses (EA), dilated intercellular spaces (DIS), and lamina propria fibrosis (LPF) contribute significantly to disease severity and progression [75,76]. The Eosinophilic Esophagitis Histologic Scoring System (EoEHSS) was developed to provide a more comprehensive evaluation of these features [76]. Collins et al. analyzed the effect of BOS on features encompassed within the EoEHSS score in a subset of patients from Dellon’s phase 2 trial [73], demonstrating that BOS significantly improved both the grade and stage of multiple histopathologic features (*p* < 0.0001) [75]. Specifically, 5 features improved in grade (PEC, BZH, EA, SL, and DIS) and 5 (EI, BZH, EA, SL, and SEA) [75]. The largest difference between the placebo and BOS groups was seen for PEC and BZH in both the grade and stage [75]. Changes in EoEHSS score moderately correlated with endoscopic improvements (R = 0.53–0.54; *p* < 0.0001), supporting its clinical relevance as a multidimensional assessment tool [75]. These data suggest that BOS not only reduces eosinophilic infiltration but also promotes broader histological healing [75]. Multiple studies have also explored BOS for long-term maintenance treatment. Dellon et al. conducted an open-label study in adult EoE patients, after having completed a phase 2 induction trial [73], assessing 24 weeks of maintenance with BOS [77]. Patients initially treated with BOS continued therapy at 2 mg once daily for 12 weeks, with optional escalation to 1.5–2 mg twice daily thereafter. At 24 weeks, 42% of initial responders maintained histologic remission, while 23% lost response. BOS was well tolerated, with minimal adverse events throughout the study period [77]. This study suggested that BOS can sustain remission in a substantial proportion of patients. Further evidence was provided by the *ORBIT2/SHP621-302* randomized withdrawal study. Patients who achieved full response after 12 weeks of BOS induction therapy were randomized to either continue BOS 2 mg twice daily or switch to placebo for 36 weeks. Relapse rates were lower in the BOS group (24%) compared to placebo (43.5%), although statistical significance was achieved only in the per-protocol analysis (*p* = 0.038) [78]. This trial demonstrated BOS’s potential to maintain long-term disease control, albeit with some risk of relapses in a subset of patients [78]. In a post hoc analysis of *ORBIT2/SHP621-302*, Dellon et al. evaluated the effects of randomized discontinuation of BOS 2.0 mg twice daily on histologic, symptomatic, and endoscopic outcomes over a 36-week period, applying a relapse definition more reflective of clinical practice (≥15 eos/HPF in at least one esophageal segment and ≥4 days of dysphagia according to the DSQ) [79]. This alternative definition, which aligns more closely with current diagnostic standards and routine clinical management of EoE, demonstrated a significantly higher relapse rate in patients withdrawing BOS compared to those who continued the treatment [79]. Furthermore, the application of LOCF imputation allowed for a more accurate assessment of treatment withdrawal effects by incorporating intervening events and medication changes during the study, thereby improving histologic, symptomatic, and endoscopic efficacy estimates at both weeks 12 and 36 compared to the original prespecified analysis [79]. These data reflected the similar earlier data on maintenance treatment with other budesonide compounds, either OVB [42] or BOT [61]. Recent pooled data showed an RR of 7.87 (95%CI, 4.19–14.77) of maintaining histological remission if treatment is continued after induction [13]. The long-term safety profile of BOS is favorable and consistent across studies. In both short-term and maintenance trials, adverse events were mild or moderate, with low rates of candidiasis (oral and esophageal) and no significant systemic complications [74,77,78]. No serious adrenal suppression was observed, with morning serum cortisol levels remaining within normal ranges. Gupta et al. further confirmed that systemic exposure to BOS remained minimal even in pediatric settings across different age groups [71]. This excellent safety profile makes BOS a particularly attractive therapeutic option for EoE, which often requires chronic, lifelong therapy. Long-term corticosteroid safety has historically raised concerns; however, BOS’s targeted esophageal delivery minimizes systemic exposure, effectively addressing these issues. Despite the growing body of evidence supporting BOS, several questions remain. First, optimal dosing regimens for long-term maintenance need further refinement, particularly balancing efficacy and safety [78]. While BOS 2 mg twice daily appears highly effective for induction, lower maintenance doses may sufficiently maintain remission in some patients while minimizing the risk of candidiasis [77,78]. Secondly, predictors of response to BOS, both clinical and molecular, are not yet well established, and biomarker research may enable better patient stratification in the future [80].

**Table 3 pharmaceutics-17-01325-t003:** Summary of main outcomes of studies on Budesonide Oral Suspension (BOS).

Study (First Author, Year)	Study Design	Intervention	Duration	Population (Age)	Primary Outcome	Key Results
Dellon et al., 2017 [73]	Phase 2, RCT, DB, PC	BOS 2 mg BID vs. placebo	12 weeks	93 patients (11–40 years)	Histological response (≤6 eos/hpf); DSQ score improvement	Histological response: 39% BOS vs. 3% placebo (*p* < 0.0001); DSQ improvement: −14.3 BOS vs. −7.5 placebo (*p* = 0.0096).
Hirano et al., 2022 [74]	Phase 3, RCT, DB, PC (*ORBIT1*)	BOS 2 mg BID vs. Placebo	12 weeks	318 patients (11–55 years)	Histologic response (≤6 eos/hpf); DSQ symptom response (≥30%)	Histological response: 53.5% BOS vs. 1% placebo (*p* < 0.001); Symptom response: 52.6% BOS vs. 39.1% Placebo (*p* = 0.024).
Gupta et al., 2015 [71]	Phase 2, RCT, DB, PC	Low, medium, high dose BOS vs. Placebo	12 weeks	71 patients (2–18 years)	Histological and symptom compound response	Responder in medium-dose: 52.6%; Responder in high-dose: 47.1%; Responder in placebo: 5.6% (*p* < 0.01). No significant difference in percentages of responders between the low-dose BOS (11.8%) and placebo groups (*p* = 0.5282).
Collins et al., 2019 [75]	Phase 2, RCT, DB, PC	BOS 2 mg BID vs. Placebo	12 weeks	87 patients (11–40 years)	EoEHSS (grade and stage) improvement	EoEHSS total scores improved for 6 of the 8 and 5 of the 8 histopathologic features for grade and stage, respectively, versus placebo. Change in EoEHSS total scores correlated moderately but significantly with change in endoscopic severity (*p* < 0.0001). The change in EoE HSS stage total score correlated weakly with the change in DSQ.
Dellon et al., 2019 [77]	Open-label extension study of a multicenter, randomized, DB, PC trial.	BOS 2 mg QD, then optional 1.5–2 mg BID	24 weeks	82 patients (11–40 years)	Histological response (≤6 eos/hpf) and change in mean peak eosinophil counts after 24 weeks	42% of patients maintained histologic response; 4% of non-responders gained response.
Dellon et al., 2022 [78]	Phase 3, RCT, DB (*ORBIT2*)	BOS 2 mg BID vs. Placebo	36 weeks	48 patients (11–55 years)	Relapse rate (≥15 eos/hpf and ≥4 dysphagia days) by 36 week	More BOS–Placebo than BOS–BOS patients relapsed over 36 weeks (43.5% vs. 24.0%; *p* = 0.131).

OLE—Open-label extension; BOS—Budesonide Oral Suspension; RCT—Randomized Controlled Trial; DB—Double-Blind; PC—Placebo-Controlled; CHR—Clinico-Histologic Remission; BID—Bis in Die; QD—Once a day; HPF—High Power Field; QoL—Quality of Life; EREFS—Eosinophilic Esophagitis Endoscopic Reference Score; EoE—Eosinophilic Esophagitis; DSQ—Dysphagia Symptom Questionnaire.

Furthermore, longer-term studies evaluating BOS’s impact on esophageal remodeling and fibrosis are warranted. Emerging data suggest that BOS may mitigate subepithelial fibrosis, but prospective studies incorporating imaging, histology, and functional esophageal metrics are needed to confirm its antifibrotic effects. BOS has recently been approved by the FDA as the first EoE-targeted STC formulation in the US.

## 5. Pharmacological Characteristics of Conventional and Novel Steroid Formulations in EoE

### 5.1. Old Formulations of Budesonide and Fluticasone (SAF/FNDS via MDI and OVB)

Initial therapeutic strategies for EoE relied on the repurposing of asthma medications, such as MDI fluticasone (both SAF or FNDS), mometasone furoate, and budesonide. Instead of being inhaled, these agents were ingested, allowing the drug to coat the esophageal mucosa and exert a localized anti-inflammatory action. In the case of fluticasone, mometasone furoate, and budesonide administered via MDI, patients release the spray (or dry powder) into the oral cavity and subsequently swallow it to achieve topical exposure within the esophagus [81]. Fluticasone propionate is a highly lipophilic inhaled corticosteroid with very low oral systemic bioavailability, estimated at approximately 1%, due to extensive first-pass hepatic metabolism. Following inhalation, drug deposition occurs in both the oropharynx and the lower respiratory tract, but only the fraction reaching the lungs contributes meaningfully to systemic absorption. The oral fraction that is swallowed undergoes near-complete inactivation during first-pass metabolism, thereby limiting systemic exposure. Absorption from the pulmonary epithelium is relatively slow compared to less lipophilic corticosteroids such as budesonide, with peak plasma concentrations typically attained around 50 min post-dose [82]. The drug exhibits a large apparent volume of distribution and high systemic clearance, reflecting rapid hepatic metabolism and tissue affinity. Fluticasone is extensively bound to plasma proteins (>90%), further influencing its pharmacokinetic behavior. The terminal elimination half-life is longer than that of budesonide, consistent with its greater lipophilicity and prolonged tissue retention [83]. Device type influences systemic bioavailability: dry-powder inhalers (DPI) generally result in lower systemic exposure than pressurized MDI, likely due to differences in pulmonary deposition patterns. Despite low overall systemic bioavailability, clinically relevant systemic effects such as cortisol suppression may occur at higher doses or with prolonged treatment, underscoring the importance of careful dose selection in pediatric and long-term therapy [32].

Mometasone furoate, on the other hand, demonstrates higher binding affinity for the corticosteroid receptor compared with fluticasone propionate or budesonide. Inhaled mometasone furoate exhibits very limited systemic absorption in healthy volunteers. The mean bioavailability of a single 400 μg dose delivered by DPI is less than 1%. Following this dose, mean plasma concentrations remain below the lower limit of quantification (0.05 μg/L); the mean area under the plasma concentration–time curve (AUC) up to the final measurable sampling point is 0.09 μg·h/L, and the peak plasma concentration is 0.05 μg/L. Systemic exposure after repeated administration of 200–800 μg/day via DPI remains minimal, as assessed by plasma concentrations of mometasone furoate. The majority of the inhaled dose is swallowed and excreted unchanged in the feces. Any absorbed fraction undergoes extensive hepatic metabolism, primarily mediated by cytochrome P450 3A4, with subsequent excretion in urine and/or bile. The systemic clearance of intravenously administered mometasone furoate 400 μg is 53.5 L/h, a value comparable to that of other inhaled corticosteroids [84].

Budesonide is a synthetic corticosteroid 1:1 mixture of two epimers (*22R* and *22S*), of which the *22R* epimer demonstrates 2- to 3-fold greater affinity for the glucocorticoid receptor and undergoes stereoselective metabolism [85]. Plasma concentrations rise in a linear fashion, with peak levels observed within 0.5–10 h following a single 9 mg oral dose. Peak plasma concentrations are typically observed within 15–45 min after inhalation. Budesonide displays a relatively short plasma half-life (≈2–3 h). On average, 60–80% of an oral capsule dose is absorbed in the ileum and colon [86]. Overall, following oral administration and absorption, budesonide displays extensive (80–90%) hepatic first-pass metabolism, primarily through cytochrome *CYP450*-3A4 and, to a lesser degree, *CYP450*-3A5, a plasma clearance of approximately 0.9–1.8 L/min in healthy adults, and relatively low systemic bioavailability [87]. Estimates vary with the route of administration, ranging from 10 to 15% for oral to 13–29% for intranasal sprays [88,89]. The drug is approximately 85–90% bound to plasma proteins, and its volume of distribution ranges between 2 and 3 L/kg, indicating widespread tissue distribution. The drug is also a substrate of P-glycoprotein, highly expressed in the intestine, which further limits systemic exposure. Its principal metabolites, *16α-hydroxyprednisolone* and *6β-hydroxybudesonide*, exhibit less than 1% of the parent compound’s glucocorticoid activity and are eliminated predominantly via the urine (60%) and feces (40%). These pharmacokinetic features, coupled with the inactivity of its metabolites, explain their strong topical anti-inflammatory efficacy and comparatively low risk of systemic corticosteroid side effects.

Importantly, because of the high first-pass metabolism, systemic bioavailability and, consequently, the risk of systemic adverse effects are significantly reduced compared with oral corticosteroids. Budesonide has been investigated mainly through the formulation of OVB or “slurry” compounds, offering a favorable therapeutic index, combining potent local anti-inflammatory activity with a relatively low risk of systemic toxicity, although growth suppression and cortisol suppression may occur in children at higher exposures or prolonged use [90]. OVB compounds (particularly the only commercialized one also for EoE in Europe, “Pulmicort Respules”, FDA/EMA approved for bronchial asthma and investigated in several RCTs for EoE) are designed to adhere to the esophageal surface, thereby prolonging mucosal contact time, enhancing local anti-inflammatory activity, and limiting the need for systemic absorption. Simeoli et al. made the first attempt to characterize the pharmacokinetics of OVB in pediatric patients with EoE associated with repaired esophageal atresia. They highlighted three major findings relevant for future evaluation of viscous formulations: (a) firstly, concentration time data revealed the presence of two distinct patient subgroups, suggesting at least two different absorption mechanisms; (b) secondly, model simulations indicated that the predominant absorption pathway may vary within the same patient across dosing occasions, likely due to physical or physiological factors such as body position or peristaltic activity influencing esophageal transit; (c) thirdly, although the 12-week treatment course was well tolerated, systemic exposure to budesonide was approximately 5-fold higher than that observed following inhaled administration in asthma [91,92]. Predicted exposure values were also 3–10 times greater than those reported for budesonide capsules in Crohn’s disease [93], and both peak concentrations (Cmax) and overall exposure were markedly elevated compared with inhaled formulations in asthma [94] and oral nebulized powders used in pediatric EoE [95]. This unexpectedly high systemic absorption is undesirable, as it could predispose to long-term steroid-related adverse effects. Nonetheless, no clinical signs of Cushing’s syndrome, diabetes, or hypertension were observed, and previously published data confirmed no significant cortisol suppression, even though the limited 12-week observation period prevented firm conclusions on long-term safety [96]. From a mechanistic standpoint, modeling identified two parallel absorption pathways of OVB: a zero-order esophageal route and a first-order gastrointestinal route. Some individuals showed predominantly zero-order absorption, others first-order, and simulations suggested that inter-occasion variability outweighed inter-individual variability. The authors suggested that this variability may be related to posture, esophageal motility, mucosal integrity, or blood flow. Notably, absorption from the esophagus circumvents hepatic first-pass metabolism, which in the case of budesonide leads to an oral bioavailability of approximately 10–15%, thereby contributing to higher systemic levels compared to gastrointestinal absorption alone [90]. It is true that esophageal absorption is characterized by a relatively longer estimated duration (about 8 h), indicating prolonged mucosal residence of the drug, similar to that observed with budesonide capsules in Crohn’s disease [93]. These findings raise the possibility that once-daily administration might achieve similar efficacy, although this was not assessed, given the limited follow-up and absence of pharmacodynamic endpoints beyond histological remission. Given the steady-state concentrations measured, patients with budesonide levels exceeding 7.5 ng/mL may face risks of cortisol suppression and reduced growth velocity. Nevertheless, since systemic exposure is not linearly proportional to dose, escalation based on height or age is not justified. Instead, lower doses (e.g., 0.25 mg twice daily or 0.5 mg once daily) may be appropriate for the overall population. Furthermore, maintaining a supine position for at least 15 min after dosing is recommended to maximize esophageal exposure and limit gastric absorption, thereby reducing unnecessary systemic exposure [91]. Pharmacokinetics of “slurry” budesonide compounds are summarized in Table 4.

### 5.2. Orally Dispersible Tablets (ODT) and Oral Suspension Formulations (OS)

#### 5.2.1. Drug Release Mechanisms

BOT (*Jorveza^®^, Budesonide 1 mg Orally Disintegrating Tablets [ODTs], EMA registration n° EU/3/13/1181, approved on 8 January 2018*) and BOS (*EOHILIA^®^, Budesonide Oral Suspension 2 mg/10 mL, FDA registration code NDC64764-105, approved on 12 February 2024*) represent complementary approaches for esophageal drug delivery. ODTs are designed to disintegrate rapidly upon contact with saliva, releasing the active pharmaceutical compound in a finely dispersed form that can dissolve locally and spread along the esophageal mucosa. In contrast, oral suspensions—particularly those developed for EoE are formulated as viscous, muco-adherent liquids that coat the mucosa and provide a sustained local presence of the drug [97]. Gupta et al. investigated the systemic pharmacokinetic profile of budesonide across varying doses of BOS in patients aged 2–18 years diagnosed with EoE [70]. Systemic exposure following administration of both low and high BOS doses was comparable across age cohorts. Increases in mean AUC0–last, AUC0–tau, and Cmax were dose-dependent, yet remained consistent between age groups at both dosing levels. Mean Tmax and terminal half-life (T1/2) did not differ substantially across either doses or age strata. No correlation was observed between clearance and volume of distribution and either body weight or body mass index (BMI), suggesting that these anthropometric measures do not influence the pharmacokinetic profile of BOS. The mean elimination half-life of BOS (3.3–3.5 h) suggests that drug accumulation is unlikely under either once-daily or twice-daily regimens. The systemic pharmacokinetics of BOS 2.0 mg have been evaluated in 47 individuals, reinforcing that the dose and volume adjustments employed in this study were appropriate for compensating for age and esophageal length. Nevertheless, Tmax in the adult cohort (~2 h) was somewhat later than that observed in the pediatric population (~1 h). By comparison, enteric-coated (EC) budesonide 9.0 mg administered once daily was studied in children aged 9–14 years with Crohn’s disease (*ENTOCORT^®^ EC (budesonide) extended-release capsules, for oral use; FDA approved 2019*). Systemic budesonide exposure was higher with EC budesonide relative to that observed in patients aged 10–18 years treated with high-dose BOS (AUC0–24: 17.78 vs. 9.68 h·ng/mL; Cmax: 2.58 vs. 0.96 ng/mL), indicating that BOS 2.0 mg administered twice daily may confer a more favorable safety profile than EC budesonide 9.0 mg once daily [96].

#### 5.2.2. Role of Excipients in Enhancing Contact Time

In BOT, excipients play a critical role in ensuring both the stability of the active compound and the effectiveness of the orally dispersible formulation. Typical excipients include fillers, such as microcrystalline cellulose, to provide an adequate tablet mass, disintegrants like crospovidone or croscarmellose sodium to facilitate rapid disintegration in the oral cavity, and binders that contribute to mechanical strength. Sweeteners and flavoring agents are frequently incorporated to improve palatability and enhance patient adherence, particularly in pediatric and adolescent populations. Lubricants such as magnesium stearate are used to optimize tablet manufacturability and prevent sticking during compression. Mucoadhesive polymers, such as carbomers or hydroxypropyl methylcellulose, promote adhesion of the formulation to the mucosal surface, thereby resisting rapid clearance caused by swallowing or peristaltic movements. Viscosity modifiers, including xanthan gum and other hydrophilic agents, increase the thickness of the suspension, prolonging mucosal contact and enabling more uniform coverage [94]. Collectively, these excipients are selected to achieve rapid dissolution, targeted mucosal adhesion, and uniform drug release, thereby maximizing local therapeutic activity in the esophagus while minimizing systemic exposure (Table 5). BOS, on the other hand, contains viscosity enhancers such as xanthan gum or hydroxypropyl methylcellulose, which increase the thickness of the suspension and prolong contact time by resisting rapid clearance during swallowing and peristalsis. Sweetening agents and flavorings are incorporated to mask the bitter taste of budesonide and improve patient acceptability, particularly in pediatric populations. Buffering agents, including citrates or bicarbonates, are often used to maintain pH stability and support drug solubility. Preservatives such as sodium benzoate or parabens may be added to ensure microbiological stability during storage. Comparison of excipients between BOT and BOS is summarized in Table 6. Comprehensively, ODT and BOS demonstrate longer residence times and greater mucosal coverage, potentially improving local drug efficacy compared to MDI swallowed and conventional oral suspensions, as shown in Table 7 [33,50,54,75].

**Table 7 pharmaceutics-17-01325-t007:** Comparison of corticosteroid delivery systems for eosinophilic esophagitis in terms of contact time and mucosal surface contact area.

Delivery System	Esophageal Mucosa Contact Time (min)	Mucosal Surface Contact Area (Relative Scale)	Notes
MDI Swallowed [33]	~3	Low	Aerosol swallowed; minimal mucosal contact
Slurry Viscous [54,56]	10–15	Moderate	Liquid flows down the esophagus; moderate contact
Orodispersible Tablets (ODT) [33]	20–30	High	Slowly dissolves in the mouth; adheres well to the mucosa
Budesonide Oral Suspension (BOS) [75]	15–20	High	Viscous liquid formulation; prolonged contact

Metered-Dose Inhaler—MDI.

#### 5.2.3. Impact of pH and Saliva on Drug Dissolution and Distribution

The physiological environment of the esophagus strongly influences the fate of orally administered corticosteroids. Salivary volume and buffering capacity affect both the disintegration of ODTs and the spreading of suspensions. Local pH variations can alter solubility profiles, with more neutral or slightly alkaline environments favoring dissolution of certain corticosteroid formulations [98].

### 5.3. ESOCAP System

The EsoCap^®^ system [38], developed for targeted drug administration to the esophagus, comprises four essential components: the rolled polymeric film, a slitted gelatin capsule, a sinker unit, and a retainer. The rolled film is enclosed within a size 00 hard gelatin capsule, in which a slot is created using a precision circular blade to enable film protrusion. To counteract buoyancy, the capsule contains a compressed sinker tablet produced from a powder mixture of calcium dihydrogen phosphate (93.6%), sodium croscarmellose (5.0%), magnesium stearate (1.0%), and iron oxide (0.4%). The sinker, manufactured via a single-punch tablet press (KP2, VEB Kombinat NAGEMA), has a diameter of 7.00 mm and a weight of 515 mg. The upper end of the polymeric film extends through the capsule slit and is secured to a polyester retainer thread of food-grade quality. This thread is connected to an application beaker filled with water, while the entire dosage form is positioned in a 3D-printed applicator linked to a drinking cup. Both the applicator and the cup are fabricated from food-grade polylactic acid (PLA) using fused deposition modeling (Ultimaker 3, Ultimaker BV, the Netherlands). The retainer serves to prevent premature unrolling of the film in the oral cavity and pharyngeal region and consists of food-safe yarn suitable for contact with mucosal tissue. The polymeric films demonstrated a smooth, homogeneous surface free from visible air inclusions. The hibiscus-based MRI contrast film showed an average thickness of 222 μm ± 2%, whereas the fluorescein sodium-loaded variant exhibited a reduced mean thickness of 120 μm ± 2%. Quantitative analysis of ten representative fluorescein sodium film samples revealed a mean drug content of 0.019 mg/mg. All values fell within 90–110% of the theoretical content, with no samples outside the 75–125% acceptance limits, thereby fulfilling the content uniformity criteria of the European Pharmacopoeia. Dissolution testing using the conventional paddle apparatus indicated that approximately 80% of the incorporated drug was released within 25 min in 500 mL of phosphate buffer at pH 7.4, with the release profile reaching a plateau after 60 min. However, this setup is not biorelevant, as it does not reflect the significantly smaller fluid volumes present in vivo. Consequently, it can be assumed that the contact time of the active substance with the esophageal mucosa is markedly longer in humans, which is expected to enhance therapeutic efficacy. Indeed, prolonged residence time on the esophageal mucosa represents a distinctive advantage of the EsoCap^®^ system. MRI investigations demonstrated that in 69% of all administrations, the polymer film remained visible for more than 15 min. In some instances, the film may have persisted longer, though imaging was terminated in accordance with study protocols. Cases where the film was not continuously visible do not necessarily indicate its absence; rather, the contrast components—derived from hydrophilic hibiscus extract—may have been gradually washed out by salivary flow from the swollen polymer matrix. This prolonged esophageal retention is particularly significant when compared with previous findings on esophageal clearance of highly viscous fluids, which were shown to be completely removed within three minutes [99]. Since residence time is a critical determinant of therapeutic success in local esophageal treatments, the mucoadhesive film technology of the EsoCap system provides clear clinical benefits.

## 6. Conclusions and Future Directions

The therapeutic landscape of EoE has undergone a profound transformation, moving from empiric symptom-based strategies to highly tailored, mechanism-driven approaches. STCs remain the backbone of therapy, but their transition from repurposed inhaler-based delivery to dedicated esophageal formulations—such as BOT/FOT and oral suspensions (BOS)—has redefined standards of care. These targeted compounds, specifically designed to maximize mucosal contact, have consistently demonstrated robust efficacy across clinical, histological, and endoscopic domains, with reassuring safety profiles even in long-term use. Yet, significant challenges remain. Relapses after treatment withdrawal are frequent, and the achievement of deep, sustained remission is limited to a minority of patients. Histological endpoints, while essential, often fail to capture the full burden of disease, particularly with regard to symptoms and fibrotic remodeling. Adherence, especially in pediatric settings, and the risk of candidiasis continue to raise practical concerns for long-term management. Biologic therapies—including agents targeting IL-4, IL-5, IL-13, and Siglec-8—are emerging as promising options for patients who are steroid-refractory and those with multi-organ eosinophilic gastrointestinal disorders. Early results, particularly with dupilumab, are encouraging, but widespread implementation will depend on accessibility, cost-effectiveness, and long-term safety data. Parallel efforts to develop reliable non-invasive biomarkers, alongside more refined histologic scoring systems such as the EoE-HSS, are critical to advancing precision medicine in EoE. Future priorities include defining predictors of sustained remission, exploring the antifibrotic potential of current and novel therapies, and evaluating the impact of early treatment initiation on long-term outcomes. Increasing awareness and timely diagnosis will be pivotal to preventing irreversible esophageal remodeling. Patient stratification in EoE using molecular signatures and non-invasive biomarkers is emerging as a key tool for personalized therapy. Promising options include: the EoE Diagnostic Panel (EDP) and transcriptomic signatures that distinguish phenotypes and predict disease activity [100]; the Esophageal String Test (EST) and other minimally invasive approaches measuring eosinophil granule proteins to monitor therapeutic response without repeated endoscopies; serum/urinary biomarkers (e.g., eosinophil-derived granule proteins) with diagnostic and predictive potential. Integration of these technologies could allow: (a) early identification of patients more likely to respond to STCs versus those requiring biologics, (b) non-endoscopic monitoring to tailor maintenance therapy, and (c) adaptive trial designs based on biomarker stratification. Large-scale validation is required to confirm predictive performance and define clinically useful thresholds for treatment decisions.

Novel, orally dispersible, and viscous steroid formulations have improved mucosal delivery, adherence, and short- to mid-term outcomes; however, long-term comparative effectiveness and safety data are still incomplete.

First, definitive head-to-head trials comparing BOT, BOS, and fluticasone ODT (and older “slurry” or swallowed-MDI approaches) are urgently needed. Such trials should include induction and maintenance phases, prespecified clinical, endoscopic, and histologic endpoints, and at least 1–2 years of follow-up to establish relative efficacy, durability of remission, and safety across patient subgroups (adult vs. pediatric; inflammatory vs. fibrostenotic phenotypes). At present, recommendations frequently rely on single-product RCTs and observational data rather than direct comparative evidence.

Second, maintenance strategies require clarification. EoE commonly relapses after therapy cessation, and although STCs demonstrate superiority over no treatment for maintenance in observational and some controlled data, optimal dosing regimens (continuous low dose vs. intermittent retreatment), minimal effective doses, and long-term safety thresholds remain uncertain. Pragmatic maintenance RCTs and real-world registry data will be necessary to inform practical, evidence-based maintenance algorithms.

Third, the role of biologics in routine care needs to be defined. Monoclonal antibodies targeting type-2 pathways (e.g., dupilumab) have demonstrated substantial histologic and symptomatic benefit in phase 3 programs, establishing biologics as an option for more severe or refractory disease and (potentially) for steroid-sparing strategies. Ongoing and future trials of other targeted agents (anti-IL-13 antibodies such as *RPC4046* and anti-Siglec-8 agents such as lirentelimab/*AK002*) show promise for additional, mechanistically distinct options that may be useful in patients who do not respond to STCs or who have mixed eosinophil/mast cell pathology. Comparative effectiveness, cost-effectiveness, and placement of biologics relative to optimized STC regimens must be defined.

Fourth, precision medicine approaches are an important future direction [101]. Transcriptomic and multi-omic studies have identified EoE-specific gene signatures and candidate biomarkers (both tissue-based and non-invasive) that correlate with disease activity and may predict response to specific therapies. Validation of noninvasive biomarkers (e.g., eosinophil granule proteins, salivary or serum signatures) could reduce dependence on repeated endoscopy and enable personalized selection between STC formulations, diet, dilation, or biologic therapy.

Fifth, safety surveillance and pediatric considerations must remain central. Long-term safety signals (HPA-axis suppression, growth effects, bone health) require standardized monitoring protocols—especially for children on prolonged or high-dose topical steroids. Registries and post-marketing studies should be prioritized to capture rare or late adverse effects and inform monitoring guidelines.

Finally, implementation and access issues (regulatory approval, reimbursement, formulation availability) are pragmatic determinants of real-world benefit. Even the most effective drug cannot improve population health unless it is affordable and acceptable to patients. Future work should therefore include health services research on the costs, adherence drivers, and interventions that improve uptake and persistence with evidence-based therapies. The main knowledge gaps and future needs are:

-Head-to-head long-term RCTs directly comparing BOT, BOS, and fluticasone ODT are required.

-Maintenance strategies need to be optimized, balancing the lowest effective dose with durable remission.

-Safety monitoring protocols (HPA-axis, growth, bone health) should be standardized in both adult and pediatric care.

-Patient-centered outcomes (adherence, QoL, preference for tablets vs. suspension) should be prioritized alongside histologic remission.

In summary, targeted esophageal STCs formulations represent a milestone in the management of EoE, offering reproducible efficacy and improved adherence. The next decade is likely to witness the integration of biologics, individualized maintenance strategies, and biomarker-driven monitoring into routine care. This convergence of innovation holds the promise not only of durable disease control but also of reshaping the natural history of EoE—transforming it from a chronic, burdensome condition into a manageable, and potentially preventable, disorder.

## Figures and Tables

**Figure 1 pharmaceutics-17-01325-f001:**
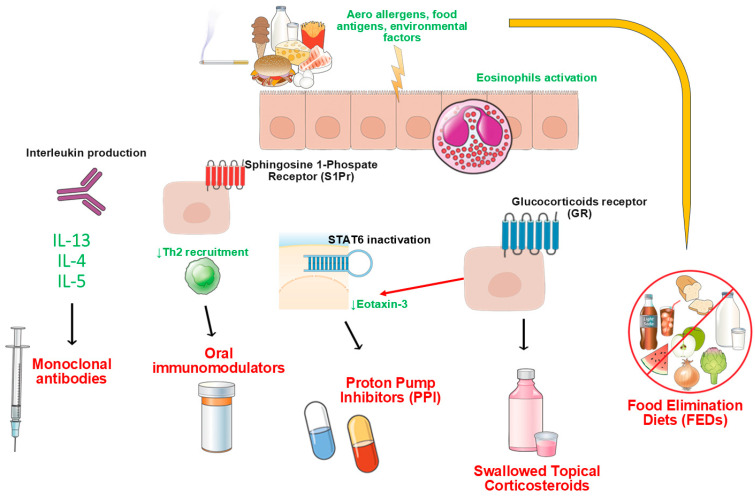
Eosinophilic Esophagitis (EoE) treatment landscape and mechanism of action of different therapies, from Proton Pump Inhibitors (PPIs) to biologics and immunomodulators. The green arrow means reduction or inhibition of the aforementioned pathways due to drug effect.

**Figure 2 pharmaceutics-17-01325-f002:**
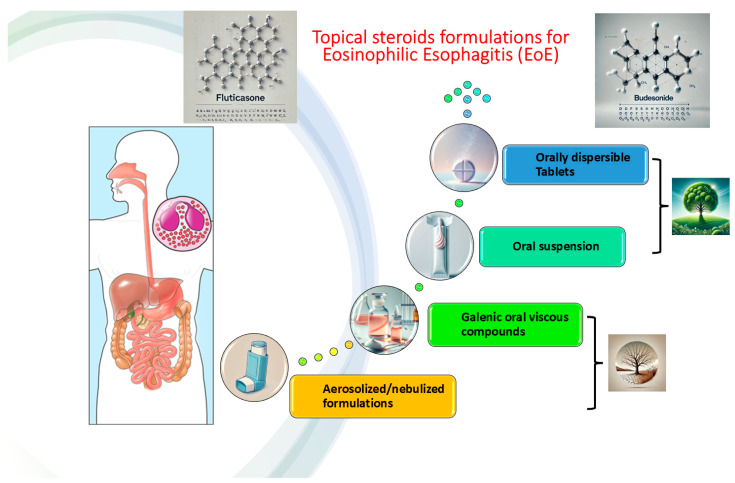
Different topical corticosteroids formulation for Eosinophilic Esophagitis (EoE), with the transition from off-label aerosolized compounds to the novel orally dispersible tablets or suspensions.

**Table 1 pharmaceutics-17-01325-t001:** Comprehensive comparative summary of off-label swallowed topical corticosteroids (olSTCs) and novel esophageal-targeted formulations for Eosinophilic Esophagitis (EoE).

	Active Principle	Formulation Type	Dosage and Posology	CHR	Efficacy in Maintenance Phase	Tolerability and Safety	Limitations
*OlSTCs*	Fluticasone	-Aerosolized swallowed with MDI -Nasal drop suspension -Home-made oral viscous solution	≤0.25 mg/day to ≥1.6 mg/day	Up to ~60–65% (dose-dependent, particularly for ≥0.8 mg/day	Decreased remission with dose reduction: 46% for fluticasone [PMID: 38284792]	Generally well tolerated; oral candidiasis reported in ~10–15%	-Non-intuitive administration methods -Home-made preparation -Non-standardized dosages -Nebulized suspensions with less mucosal adherence compared to viscous slurry compounds
	Budesonide	-OVB (budesonide nebulizer suspension + sucralose or cellulose) -Aerosolized swallowed with MDI	-OVB 1–2 mg/day (0.25 mg BID in maintenance)	-~72–80% for OVB (2 mg/day); no added benefit above 4 mg/day	-OVB maintained remission (>65% overall and over discontinuation registered a significant increase in PEC in placebo, *p* = 0.024) -No difference in High Vs. Lower dosage but early relapses at low dose		
	Mometasone	-Aerosolized mometasone with MDI	up to 1500 μg/day	-Median eos HPF change from baseline (−50, *p* < 0.001) -significant improvement in dysphagia score, not QoL	-No data available	-No major adverse events reported	-10× lower bioavailability compared to fluticasone and >300× lower than budesonide -No data on histologic and endoscopic outcomes
*Novel STCs*	Fluticasone	-FOT (*APT-1011*)(tablets to be merged with saliva and swallowed)	-3 mg BID -3 mg HS -1.5 mg BID -1.5 mg HS	->65% up to 100% of histologic remission -Significant improvement in dysphagia scores for all dosages -Good response also for EREFS in fibrostenotic patients	-Histological response maintained (up to 84%) at 52 weeks with 1.5 mg BID. Lower (30%) for 1.5 mg QD	-Safe profile overall with candidiasis as the most frequent event (12–16%, usually mild)	-Enhanced esophageal targeting -Favorable tolerability -High patient adherence due to the ease of administration -No need for compounding -Higher patient satisfaction with OVB
	Budesonide	-BOT (tablets to be merged with saliva and swallowed) -BOS (syrup-like consistency with two viscosity-modifying agents)	-BOT: 1–2 mg/day (0.5–1 mg BID for maintenance) -BOS: 2 mg BID (2 mg QD for maintenance)	-From 70% to 100% CHR with BOT (2 mg/day); OR 18.9 for remission (*p* < 0.001) in EoE -BOS 45–50% of histological response in pediatric (clinical improvement non-significant) -BOS in adults showed significant CHR (*p* < 0.0001 and *p* = 0.0096)	-BOT reported CHR up to 75% at 52 weeks and >80% at 96 weeks -Relapse rates were lower in the BOS group (24%) compared to placebo (43.5%)		
	Mometasone	-Mometasone (*ESO-101*)		-Histological remission in 48% compared to 0% in placebo -Endoscopic improvement with *ESO-101* compared to placebo but not for clinical symptoms	-No trials available	No adverse event reported	

OlSTCs—Old STCs formulations; STCs—Swallowed Topical Corticosteroids; OVB—Oral Viscous Budesonide; BOT—Budesonide Orally Dispersible Tablets; BOS—Budesonide Oral Suspension; RCT—Randomized Controlled Trial; CHR—Clinico-Histologic Remission; BID—Bis in Die; QD—Once a day; HS—Bedtime; FOT—Fluticasone Orally Dispersible Tablets; MDI—Metered-Dose Inhaler; HPF—High Power Field; QoL—Quality of Life; EREFS—Eosinophilic Esophagitis Endoscopic Reference Score; EoE—Eosinophilic Esophagitis.

**Table 4 pharmaceutics-17-01325-t004:** Pharmacokinetic characteristics of oral viscous budesonide (OVB) compounds.

	Findings	Clinical Implications
**Absorption mechanisms**	Two parallel absorption pathways: zero-order esophageal absorption and first-order gastrointestinal absorption	Explains variability in drug exposure and highlights the role of esophageal mucosal residence time in determining efficacy and safety.
**Systemic exposure**	Considerably higher than with inhaled budesonide (≈5×) and capsules for Crohn’s disease (≈3–10×). AUC and C_max_ are also greater than those of the oral suspension for EoE.	Risk of unwanted systemic effects (e.g., cortisol suppression, growth impairment).
**Variability**	High inter-individual and inter-occasion variability, not explained by age, weight, or dose.	Suggests influence of posture, motility, and mucosal status. Dose adjustments may not solve variability.
**Mucosal residence**	Zero-order absorption lasted ≈ 8 h, suggesting prolonged esophageal contact.	Once-daily dosing may be sufficient to maintain therapeutic effect.
**Safety**	Short-term and even long-term treatment is well tolerated by both for pediatric and adult populations	A longer follow-up is needed to confirm safety in chronic use.
**Dose rationale**	Exposure is not linearly related to dose; higher doses are not justified. Lower doses (0.25–0.5 mg/day) and supine administration are recommended.	Supports re-evaluation of standard dosing strategies to minimize systemic absorption.

AUC—Area Under the Curve Cmax Maximum Concentration.

**Table 5 pharmaceutics-17-01325-t005:** Excipients used in Budesonide Orally Dispersible/Orally Disintegrating Tablets.

Function	Excipients	Possible Contributions to the Formulation
Effervescence/disintegration/pH buffer	*Sodium acid citrate, Disodium (or sodium di-) hydrogen citrate, Sodium hydrogen carbonate*	These agents react in the presence of saliva to produce an effervescent effect, helping the tablet to disintegrate/dissolve rapidly in the mouth. The citrate/bicarbonate system creates mild fizz and promotes saliva production, which enhances the dispersal of the active ingredient.
Plasticizer/surfactant/wetting	*Docusate sodium*	Helps with wetting, perhaps aiding uniform distribution of saliva and aiding disintegration.
Polymer binder/film forming	*Povidone (K25)*	Acts as a binder to help tablet cohesion; it may also affect the dissolution profile.
Filler/diluent/bulking agent	*Mannitol (E421)*	Used to give mass/volume to the tablet; often gives a pleasant mouthfeel; relatively water-soluble, so it contributes to dissolution.
Lubricant	*Magnesium stearate*	To reduce friction during tablet manufacture, avoid sticking in the tablet press; this ensures the tablets can be produced with uniformity.
Polyethylene glycol (PEG)/Macrogol	*Macrogol 6000*	Acts as a PEG/disperser; helps in dissolution/disintegration; can enhance the wetting properties of the tablet.

**Table 6 pharmaceutics-17-01325-t006:** Comparison of Excipients in Budesonide Orally Dispersible Tablets (BOT) vs. Budesonide Oral Suspension (BOS).

Function	BOS—Suspension	BOT—Orally Dispersible Tablets
*Vehicle/Base*	Purified water	Mannitol (filler/diluent)
*Sweeteners and Flavors*	Acesulfame K, Magnasweet^®^, dextrose, maltodextrin, cherry flavor	Sucralose
*Viscosity/Mucoadhesion*	Avicel^®^ RC-591 (microcrystalline cellulose + Na-CMC), glycerine	Povidone K25, Macrogol 6000, Docusate sodium
*Disintegration*	no needed	CO2 gas disintegrant
Stabilizer *(antioxidants/* *chelators)*	Ascorbic acid, sodium ascorbate, disodium EDTA	Not reported
*Lubricants (manufacturing)*	Not required (liquid formulation)	Magnesium stearate
*Buffering/pH control*	Citric acid + sodium citrate	Sodium dihydrogen citrate

Data extracted from European Medicines Agency (EMA). Jorveza: public assessment report.

## Data Availability

No new data were generated or analyzed in support of this research. No new data were created or analyzed in this study. Data sharing is not applicable to this article.

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
