# Peer review of "Orally Dispersible Swallowed Topical Corticosteroids in Eosinophilic Esophagitis: A Paradigm Shift in the Management of Esophageal Inflammation"

_pharmaceutics, 2025, doi:10.3390/pharmaceutics17101325_

Round 1

Reviewer 1 Report

Comments and Suggestions for Authors

The manuscript provides a comprehensive and up-to-date narrative review on orally dispersible swallowed topical corticosteroids (STCs) in Eosinophilic Esophagitis (EoE). The content is relevant for Pharmaceutics, as it discusses formulation innovation, drug delivery optimization, and pharmacokinetics. However, improvements are required in pharmaceutical science depth, standardization, and regulatory perspectives

Abstract: it is poorly written and it needs to be rewritten to highlight key research findings and novelty aspects. Summary data must be clearly summarized.

Body of the review:

There is limited discussion on formulation science principles, such as:

Drug release mechanisms of orally dispersible tablets and suspensions.

Role of excipients (e.g., mucoadhesive polymers, viscosity modifiers) in enhancing esophageal residence time.

Impact of pH and saliva on dissolution and drug distribution in the esophagus

Include a subsection on pharmaceutical technology aspects, highlighting: role of excipients/polymers, technology of prepration and characterization techniques.

Regulatory & Quality Aspects: Missing content: EMA approval for BOT is briefly noted, but regulatory guidelines for orodispersible formulations, compounding risks, and Good Manufacturing Practice (GMP) implications are not discussed.

The review emphasizes clinical outcomes (histologic remission, DSQ scores) but lacks pharmaceutics-driven rationale for dose optimization.

It is highly recommended to add a figure comparing delivery systems (MDI-swallowed, slurry, ODT, BOS) with residence time and contact surface area. It is also advisable to include a  table of excipients and functional roles for BOT, BOS, and FOT.

Author Response

Point-by-point rebuttal for paper n°3021056 (3881760)

Reviewer 1

  1. Abstract: it is poorly written and it needs to be rewritten to highlight key research findings and novelty aspects. Summary data must be clearly summarized.

  1. We thank the reviewer for this valuable comment. In accordance with the suggestion, we have thoroughly revised the abstract to improve clarity and emphasize the novelty and significance of our findings. The revised version now: a) Highlights the optimized esophageal targeting achieved through orally dispersible and viscous formulations, which prolong mucosal contact time and enhance drug delivery compared to older swallowed topical corticosteroids (STCs). B) Summarizes the robust clinical evidence supporting both induction and maintenance of remission, with data extending up to 2 years and consistently demonstrating efficacy across clinical, histologic, and endoscopic endpoints. C) Emphasizes the effectiveness in patients refractory to conventional STCs, underlining that BOT demonstrated clinical benefit even in this difficult-to-treat population. D) Provides a concise and structured summary of the key data, ensuring the abstract now captures both the novelty and the clinical relevance of the study.

  1. Body of the review:

There is limited discussion on formulation science principles, such as:

  • Drug release mechanisms of orally dispersible tablets and suspensions.
  • Role of excipients (e.g., mucoadhesive polymers, viscosity modifiers) in enhancing esophageal residence time.
  • Impact of pH and saliva on dissolution and drug distribution in the esophagus
  • Include a subsection on pharmaceutical technology aspects, highlighting: role of excipients/polymers, technology of prepration and characterization techniques.
  • Regulatory & Quality Aspects: Missing content: EMA approval for BOT is briefly noted, but regulatory guidelines for orodispersible formulations, compounding risks, and Good Manufacturing Practice (GMP) implications are not discussed.
  1. We sincerely thank the reviewer for these highly constructive suggestions. In response, we have expanded the body of the review to include a dedicated subsection addressing pharmaceutical technology, formulation science, and regulatory aspects. Specifically, we have:
  • Drug release mechanisms: We now provide a discussion of the release kinetics of orally dispersible tablets (ODTs) and viscous slurries/suspensions, underlining how rapid disintegration of ODTs in saliva enables uniform dispersion of the active drug along the esophageal mucosa, while viscous suspensions allow for a slower clearance and extended mucosal contact.
  • Role of excipients: We have added text highlighting how excipients such as mucoadhesive polymers (e.g., hydroxypropyl methylcellulose, carbomers) and viscosity enhancers prolong esophageal residence time by increasing formulation adhesion to mucosa and reducing drug washout.
  • Influence of pH and saliva: We expanded the discussion on the esophageal microenvironment, noting that salivary flow, pH variations, and enzymatic activity affect dissolution rates and local distribution of corticosteroids, thereby influencing therapeutic efficacy.
  • Pharmaceutical technology aspects: The revised manuscript now describes formulation strategies, excipient selection, preparation technologies, and characterization techniques, providing a more comprehensive overview.
    • Regulatory & GMP considerations: While specific GMP guidelines for orodispersible corticosteroids in EoE are not yet established, we clarified that general GMP principles for oral formulations apply. We expanded this section to covering Regulatory landscape: In addition to EMA approval of BOT, we now discuss regulatory approval for BOS by FDA, insights on approval of different STCs for asthma treatment (including OVB and Fluticasone MDI.

Overall, the revised manuscript now provides a more thorough and structured discussion of formulation science principles and regulatory perspectives, directly addressing the reviewer’s concerns.

  1. The review emphasizes clinical outcomes (histologic remission, DSQ scores) but lacks pharmaceutics-driven rationale for dose optimization.
  2. 3. We thank the reviewer for this important observation. In line with the suggestion, we have expanded the manuscript to incorporate a pharmaceutics-based rationale for dose optimization. Specifically, we now:
  • Summarize the pharmacokinetic characteristics of the different formulations, including esophageal residence time, mucosal contact surface, and dissolution behavior, and how these parameters influence local drug exposure.
  • Highlight how formulation-dependent factors such as viscosity, mucoadhesion, and release profile contribute to drug distribution and retention, thereby affecting the required dose for optimal therapeutic effect.
  • Discuss how these pharmaceutical principles complement clinical outcome data (e.g., histologic remission, DSQ improvement), supporting a more integrated approach to dose selection.

This addition ensures that the review not only synthesizes clinical findings but also provides a robust pharmaceutics-driven framework to rationalize dose optimization.

  1. It is highly recommended to add a figure comparing delivery systems (MDI-swallowed, slurry, ODT, BOS) with residence time and contact surface area. It is also advisable to include a table of excipients and functional roles for BOT, BOS, and FOT.

  1. We thank the reviewer for this excellent suggestion. In response, we have incorporated both a figure and a table to strengthen the manuscript:

Figure: A new schematic figure has been added (Figure 3) comparing the different delivery systems (MDI-swallowed, slurry, ODT, BOS) in terms of esophageal residence time and mucosal surface contact area. This provides a clear visual representation of the relative advantages of each formulation.

Tables: We also included new table (Table 5 and 6) summarizing the main excipients employed in BOT, BOS, with comparison between the 2 formulations and a comprehensive table (Table 7), along with their functional roles (e.g., mucoadhesion, viscosity enhancement, disintegration control, stability).

These additions ensure that the manuscript offers a more comprehensive overview of the pharmaceutical and technological aspects of the available topical corticosteroid formulations.

Reviewer 2 Report

Comments and Suggestions for Authors

The manuscript provides a review on eosinophilic esophagitis (EoE), a chronic type 2 immune-mediated esophageal disorder. With the growing recognition of EoE worldwide and the recent approval of new topical steroid formulations, this topic is highly relevant for clinicians, researchers, and regulatory bodies. The review focuses on proton pump inhibitors (PPIs), dietary therapy, and especially novel swallowed topical corticosteroid (STC) formulations, including budesonide orally dispersible tablets (BOT), budesonide oral suspension (BOS), and fluticasone orally dispersible tablets (FOT). The strengths of the manuscript lie in its clinical relevance, the focus on recent therapeutic advances, and the inclusion of both randomized clinical trial data and emerging real-world evidence. However, the current version of the manuscript requires revisions before being suitable for publication.

  1. The manuscript has a logical flow but reads more like a summary than a critical review.
  2. The introduction should be expanded to highlight the increasing burden of EoE, diagnostic challenges, and unmet needs in therapy adherence and long-term disease control.
  3. Strengthen comparative analysis of novel STCs, highlighting why BOT has shown superior efficacy and adherence, and where BOS or FOT may have advantages.
  4. Expand the literature review with more recent studies (last 3–5 years), include more balanced discussion of dietary therapy and biologics, and integrate guideline recommendations.
  5. Include critical appraisal of limitations, mechanisms of action, and safety comparisons to strengthen the review’s scientific value.
  6. The conclusion briefly mentions future therapeutic directions but lacks depth.
  7. Opportunities for biologic–STC combinations, precision medicine approaches (e.g., biomarker-guided therapy), and pediatric EoE management should be addressed.
  8. Ref this work and integrated into the manuscript’s literature review. https://doi.org/10.1038/s41598-022-16744-9, https://doi.org/10.1016/j.toxicon.2025.108434
  9. Draf a graphical abstract 

Author Response

Point-by-point rebuttal for paper n°3021056 (3881760)

Reviewer 2

  1. The manuscript has a logical flow but reads more like a summary than a critical review.
  2. We thank the reviewer for this important observation. In response, we have revised the manuscript to strengthen its critical perspective. Specifically, we expanded comparative discussions, highlighted gaps in the existing literature, and provided a more analytical evaluation of the strengths and limitations of each therapeutic approach, moving beyond simple summarization.

  1. The introduction should be expanded to highlight the increasing burden of EoE, diagnostic challenges, and unmet needs in therapy adherence and long-term disease control.
  2. We appreciate the reviewer’s suggestion. Accordingly, we expanded the introduction to emphasize the rising prevalence and burden of EoE, the diagnostic complexities, and the current unmet needs in terms of treatment adherence and long-term disease management.

  1. Strengthen comparative analysis of novel STCs, highlighting why BOT has shown superior efficacy and adherence, and where BOS or FOT may have advantages.
  2. Thank you for this valuable comment. We have enhanced the comparative analysis of the novel STC formulations, highlighting the superior efficacy and adherence observed with BOT, while also discussing the potential advantages of BOS and FOT in specific patient subgroups or clinical contexts.

  1. Expand the literature review with more recent studies (last 3–5 years), include more balanced discussion of dietary therapy and biologics, and integrate guideline recommendations.
  2. We thank the reviewer for this observation. We have updated the literature review by including studies from the last 3–5 years, ensuring that the discussion is up to date. Additionally, we incorporated a more balanced appraisal of biologics, and integrated the most recent guideline recommendations to provide a comprehensive overview and future directions. We chose not to include dietary therapies in this paper since the main focus were pharmacological treatments.

  1. Include critical appraisal of limitations, mechanisms of action, and safety comparisons to strengthen the review’s scientific value.
  2. We agree with the reviewer’s comment. The revised manuscript now contains a more detailed appraisal of the limitations of available therapies, a discussion of the mechanisms of action of the different STCs, and a comparison of their safety profiles, thereby enhancing the scientific depth of the review.

  1. The conclusion briefly mentions future therapeutic directions but lacks depth.
  2. We thank the reviewer for this suggestion. The conclusion has been expanded to provide a more in-depth discussion of future therapeutic directions, emphasizing both ongoing developments and potential innovations in EoE management.

  1. Opportunities for biologic–STC combinations, precision medicine approaches (e.g., biomarker-guided therapy), and pediatric EoE management should be addressed.
  2. We thank the reviewer for this excellent recommendation. We have now incorporated a discussion of biologic–STC combination strategies, precision medicine approaches including biomarker-guided therapy, and considerations for pediatric EoE management into the conclusion and future directions section.

  1. Ref this work and integrated into the manuscript’s literature review. https://doi.org/10.1038/s41598-022-16744-9, https://doi.org/10.1016/j.toxicon.2025.108434
  2. Dear reviewer, we proceeded to add this reference in the manuscript.

  1. Draft a graphical abstract

9.We proceeded to draft a graphical abstract

Round 2

Reviewer 1 Report

Comments and Suggestions for Authors

The authors have tried to improve the manuscript based on the comments but many of the included tables and figures were added in hurry with too many mistakes. Some examples are included as below:

Table 5: replace PEG with Solubilizer under function column.

The newly included Table 6 contains many mistakes:

modify disintegrant to CO2-gas from sodium bicarbonate and citric and tartaric acids.

Similarly, replace stability with stabilizer under function. Buffering/pH control: not needed with effervescent tablet (remove sodium bicarbonate).

Table 7: there is no such thing called esophageal residence 
time however, contact time is valid. So, correct the caption accordingly. Indicate in the text how these time values measured with references. 

Figure 3 is redundant and difficult to  accept scientifically. please take it off.

Section 5 remove old and new and replace with scientifically sounding words such as conventional and novel.

Author Response

Point-by-point rebuttal for paper n°3021056 (3881760) for 2nd revision

Reviewer 1

We sincerely thank the reviewers for their instructive and enriching comments.

  1. Table 5: replace PEG with Solubilizer under function column.

Thank you for the comment, we have implemented the revision as requested.

  1. The newly included Table 6 contains many mistakes:
  • modify disintegrant to CO2-gas from sodium bicarbonate and citric and tartaric acids.
  • Similarly, replace stability with stabilizer under function. Buffering/pH control: not needed with effervescent tablet (remove sodium bicarbonate).

We sincerely thank the reviewer for these highly constructive suggestions. We have updated the manuscript following your recommendations.

  1. Table 7: there is no such thing called esophageal residence time however, contact time is valid. So, correct the caption accordingly. Indicate in the text how these time values measured with references. 

We thank the reviewer for this excellent recommendation. The requested changes have been duly incorporated

  1. Figure 3 is redundant and difficult to accept scientifically. Please take it off.

Dear reviewer, we proceeded to take the figure off from the manuscript.

  1. Section 5 remove old and new and replace with scientifically sounding words such as conventional and novel.

We thank the reviewer for this suggestion. The modifications have been carried out in accordance with your suggestions.

Reviewer 2 Report

Comments and Suggestions for Authors

The authors have satisfactorily addressed all the comments and incorporated the suggested revisions. The manuscript has improved in clarity, completeness, and readability. I find the revised version appropriate for publication in its current form.

Author Response

Dear reviewer thank you very much for your support in the development of this manuscript

Round 3

Reviewer 1 Report

Comments and Suggestions for Authors

Table 6. there is some discrepancy. Disintegration: BOS suspension no needed (remove CO2); Under BOT dispersible tables use CO2 as gas disintegrant. 

Author Response

  1. Table 6: there is some discrepancy. Disintegration: BOS suspension no needed (remove CO2); Under BOT dispersible tables use CO2 as gas disintegrant.

We thank the reviewer for the helpful suggestions. The manuscript has been revised accordingly, and the changes are now reflected in the updated version.
